# A Combined Transcriptomic and Proteomic Analysis of Monkeypox Virus A23 Protein on HEK293T Cells

**DOI:** 10.3390/ijms25168678

**Published:** 2024-08-08

**Authors:** Yihao Wang, Yihan Li, Mingzhi Li, Keyi Wang, Jiaqi Xiong, Ting Wang, Yu Wang, Yunli Guo, Lingbao Kong, Meifeng Li

**Affiliations:** 1Institute of Pathogenic Microorganism, Jiangxi Agricultural University, Nanchang 330000, China; wangisbalance@outlook.com (Y.W.); liyihan@jxau.edu.cn (Y.L.); 18537677196@163.com (M.L.); kywang0407@163.com (K.W.); jackieqix@163.com (J.X.); tingwang@jxau.edu.cn (T.W.); wangyu@jxau.edu.cn (Y.W.); 15079146962@139.com (Y.G.); 2Nanchang City Key Laboratory of Animal Virus and Genetic Engineering, Nanchang 330000, China; 3College of Bioscience and Engineering, Jiangxi Agricultural University, Nanchang 330000, China

**Keywords:** transcriptome, proteomic, mpox virus, A23R, innate immune signaling pathway

## Abstract

Monkeypox virus (MPXV) is a cross-kingdom pathogen infecting both humans and wildlife, which poses a significant health risk to the public. Although MPXV attracts broad attention, there is a lack of adequate studies to elucidate pathogenic mechanisms associated with viral infections. In this study, a high-throughput RNA sequencing (RNA-seq) and liquid chromatography–tandem mass spectrometry (LC-MS/MS) approach was used to explore the transcriptional and metabolic responses of MPXV A23 protein to HEK293T cells. The protein–protein interactions and signaling pathways were conducted by GO and KEGG analyses. The localization of A23 protein in HEK293T cells was detected by immunofluorescence. A total of 648 differentially expressed genes (DEGs) were identified in cells by RNA-Seq, including 314 upregulated genes and 334 downregulated genes. Additionally, liquid chromatography–tandem mass spectrometry (LC-MS/MS) detected 115 cellular proteins that interact with the A23 proteins. Transcriptomic sequencing analysis revealed that transfection of MPXV A23 protein modulated genes primarily associated with cellular apoptosis and DNA damage repair. Proteomic analysis indicated that this protein primarily interacted with host ribosomal proteins and histones. Following the identification of the nuclear localization sequence RKKR within the A23 protein, a truncated mutant A23_ΔRKKR_ was constructed to investigate the subcellular localization of A23 protein. The wild-type A23 protein exhibits a significantly higher nuclear-to-cytoplasmic ratio, exceeding 1.5, in contrast to the mutant A23_ΔRKKR_, which has a ratio of approximately 1. Immunofluorescence assays showed that the A23 protein was mainly localized in the nucleus. The integration of transcriptomics and proteomics analysis provides a comprehensive understanding of the interaction between MPXV A23 protein and the host. Our findings highlight the potential role of this enzyme in suppressing host antiviral immune responses and modulating host gene expression.

## 1. Introduction

Monkeypox virus (MPXV) is the causative agent of Monkeypox (mpox), a zoonotic disease that causes a smallpox-like rash [1]. MPXV was first described in 1958 among macaques shipped from Singapore to Denmark [2]. In 1970, the first human case of mpox was reported in a 9-month-old boy in the Democratic Republic of the Congo [3]. Subsequently, sporadic outbreaks of mpox occurred in Africa, but with limited international attention. Between 2003 and 2022, there was an increase in the number of mpox cases reported outside the epidemic countries, and the risk of human-to-human transmission of the virus remains a constant threat [4]. However, since May 2022, there has been a global outbreak of mpox that has spread among large populations, which promoted the World Health Organization (WHO) to declare the mpox epidemic “a public health emergency of international concern” in July 2022 [5]. As of December 2022, the number of reported mpox patients worldwide has exceeded 79,000 and the virus has spread to 6 continents, leaving only Antarctica with no reported cases [6].

MPXV is a double-stranded DNA virus belonging to the Orthopoxvirus genus of Poxviridae, which also includes variola, vaccinia, and cowpox viruses [7]. Mpox exhibits clinical similarities to smallpox, but with less severe symptoms. Moreover, the administration of the smallpox vaccine provides a cross-protective impact against MPXV and exhibits efficacy in mitigating clinical manifestations [7]. One proposed hypothesis suggests that the rise in mpox cases can be linked to the widespread cessation of smallpox vaccination, resulting in an increased susceptibility of humans to the MPXV. Consequently, the virus has been able to evolve advantageous immune evasion strategies in response to selective pressures [8]. Based on epidemiological studies, the 2022 outbreak was attributed to the MPXV of linear group B.1 among the West African clade (MPXV Clade 3), which encompasses at least 46 single-nucleotide polymorphisms (SNP) specific for this lineage [9]. The observed rate of mutations in MPXV is significantly higher than the anticipated natural rate of Orthopoxvirus, implying that the dissemination and emergence of MPXV in novel regions have accelerated its mutational evolution. In contrast to the variola virus, MPXV has a wider range of hosts, and the persistent inter-species transmission may facilitate its capacity for host switching and efficiency in human transmission [8]. Despite the successful eradication of smallpox, knowledge gaps still exist regarding viral pathogenesis and the host’s immune responses to poxvirus infections. The outbreaks of mpox serve as a reminder that the ongoing emergence and adaptation of zoonotic poxviruses pose a persistent threat to global health.

As with other poxviruses, the replication and expression of MPXV is a multifaceted process that occurs exclusively in the host cytoplasm. The MPXV genome comprises over 190 genes that possess the capacity to encode its own transcription machinery. Nonetheless, the virus relies on host cells for translation, implying that the host cellular stress responses have a direct impact on the viral reproductive efficacy [10]. However, the investigation on the interplay between MPXV and host is significantly limited, with the majority of current studies relying on research conducted on vaccinia virus (VACV) and other poxviruses.

The MPXV-encoded proteins are categorized into three distinct groups: (1) entry proteins, which facilitate the entry of MPXV into host cells through receptor binding and membrane fusion, including M1R, E8L, and H3L; (2) exit proteins, which aid in the release of MPXV copies from host cells, including A38R, C23R, and C18L; and (3) immunomodulatory proteins, which are crucial for the modulation of host cell functions and immune responses, including J2L, A47L, and A41L (https://viralzone.expasy.org/9976, accessed on 28 July 2024).The MPXV A23 protein is a poxvirus A22 resolvase belonging to the RNase H superfamily and plays a pivotal role in viral replication by resolving Holliday junctions (HJs) that emerge at the junctions of inverted repeats in a linear genomic DNA concatemer [11,12]. HJ resolvases have been identified in a variety of organisms and are typically categorized into two distinct functional groups [13]. The first group, derived from bacteriophages, displays a diverse range of activities on branched DNA substrates and primarily functions in double-strand break repair and DNA recombination prior to packaging [14,15]. In contrast, the second functional group, exemplified by *Escherichia coli* RuvC, exhibits specificity towards HJ structures exclusively. Despite belonging to the RuvC family, the A22 protein exhibits a broader substrate specificity for DNA branching, more resembling the bacteriophage enzymes [16]. This distinction suggests an evolutionary adaptation in viral resolvases to efficiently process diverse branched intermediates, which may have deleterious effects on their cellular counterparts [17]. The standard database search found a significant similarity between the A22R orthologs of poxviruses, highlighting their importance in viral biology.

The conservation of HJ resolvase in poxviruses and its essential role in viral replication make it a promising therapeutic target due to the absence of analogous enzymes in mammalian cells [18]. The HIV integrase enzyme, another member of the RNase H superfamily, has been a focus of drug development efforts, potentially serving as a viable foundation for the development of poxvirus antivirals [19]. Nevertheless, a comprehensive understanding of its mechanism remains elusive. In this study, we employed transcriptome analysis to investigate the impact of the A23 protein on gene expression in HEK293T host cells. Additionally, we utilized mass spectrometry to identify proteins that potentially interact with host cell proteins. The objective was to enhance our understanding of the effects of the A23 protein on host cell proteins, thereby contributing to a broader comprehension of MPXV pathogenesis.

## 2. Results

### 2.1. Plasmids Construction and Western Blot Analyses of A23 Protein

The pCAGGS-HA-A23R plasmid was constructed and verified by DNA agarose gel electrophoresis. The “GAATTC” and “CTCGAG” sequences at both ends of A23R represent *EcoR*I and *Xho*I cleavage sites (Figure 1A). A double digestion assay was used to verify the pCAGGS-HA-A23R plasmid, which showed two prospective bands; one band is the pCAGGS vector (about 5000bp) and the other is the A23R gene (about 500 bp) in gel (Figure 1B). Then, the pCAGGS-HA-A23R plasmid was transfected into HEK293T cells and the A23 protein was detected by the HA-tag antibody. The result of the Western blot showed that A23 protein (a 25 kD band) can be detected, which suggested that pCAGGS-HA-A23R plasmid was successfully expressed in HEK293T cells (Figure 1C). Partial samples of the Co-IP and the WCL after anti-HA incubate (flowthrough) were subjected to Western blot analyses. The results indicate that most of the A23 protein is adsorbed by the anti-HA agarose conjugate (Figure 1D).

### 2.2. Transcriptome Sequencing Evaluation

The Illumina sequencing was performed on pCAGGS-HA-A23R/control vector transfected HEK293T cells. After raw data filtration, error rate examination, and GC content distribution, at least 6.47 Gb of clean reads from each sample were obtained with GC% > 49.13%, Q20% > 96.11%, Q30% > 90.02%, and the error rate < 0.03% (Table 1). For further analysis, high-quality clean reads were well mapped onto the human reference genome (GRCh38) using Hisat2, which showed an average mapping rate of 94.76%, including 92.28% unique mapping reads and 2.48% multiple mapping reads. In summary, these results indicated that the transcriptome sequencing data were appropriate and reliable for further analysis.

### 2.3. Identification of DEGs

When compared with cells transfected with empty vectors, a total of 648 DEGs were screened in HEK293T cells transfected with pCAGGS-HA-A23R, of which 314 genes were upregulated, and 334 genes were downregulated. Volcano plot and hierarchical clustering analysis provided an overview of gene expression differences (Figure 2). Notably, a considerable portion of genes associated with cell apoptosis (e.g., *LIG4*, *BIRC3*, and *SMAC*) and DNA damage repair (e.g., *RAD9B*, *LIG4*, and *SGO2*), were significantly differently expressed.

Twelve DEGs (*IL9R*, *PLA2G4C*, *MAFA*, *CYP2E1*, *H3C1*, *H2BC17*, *HLA*-*DPB1*, *LIG4*, *BIRC3*, *SMAC*, *RAD9*, and *SGO2*) were selected via qRT-PCR analysis to further support the credibility of RNA-Seq (Figure 3, Table 2).

### 2.4. Functional and Pathway Enrichment Analysis of DEGs

GO analysis was used to analyze the potential biological functions of DEGs. The results of GO analysis showed that DEGs were categorized into three groups, including biological process (BP, 14 GO terms), cellular component (CC, 12 GO terms), and molecular function (MF, 11 GO terms). The top 30 enriched GO terms were shown in Figure 4A. Biological processes including epigenetic regulation of gene expression (GO: 0040029), telomere organization (GO: 0032200), and DNA replication-dependent nucleosome assembly (GO: 0006335) were highly enriched. In the cellular component category, nuclear chromosome (GO: 0000228) and nucleosome (GO: 0000786) were the main enriched terms. In the category of molecular function, genes were primarily related to structural constituents of chromatin (GO: 0030527). The KEGG pathway enrichment analysis showed that transcriptional misregulation in cancer, arachidonic acid metabolism, and linoleic acid metabolism were statistically significantly enriched (Figure 4B).

### 2.5. Interaction of MPXV A23R

To further explore the mechanism of the poxvirus resolvase, LC-MS/MS analysis was applied to screen the interaction proteins of MPXV A23R. Heterologous expression of MPXV A23R in HEK293 cells was extracted for Co-IP with anti-HA beads, and Western blotting was conducted to verify the protein expression (Figure 1D). The Co-IP products were subjected to LC-MS/MS analysis, and the details are presented in Table 3. The LC-MS/MS analysis revealed a significant number of ribosomal proteins (e.g., RPL10A, RPS26, and RPS28). RNA polymerases (e.g., POLR2H, POLR2G) can interact with MPXV A23 protein, implying the potential regulatory influence of this protein on host gene translation, which may facilitate viral replication. Additionally, we also observed a considerable number of histones (e.g., H2AC4, H2AZ1, and H2AX) which suggested that MPXV A23 protein may play an important role in regulating the biological function of host proteins in the nucleus. GO enrichment and KEGG analysis were further carried out to explain the biological effects of the identified proteins. The GO enrichment analysis showed that telomere organization (GO:0032200) in biological processes, the nuclear chromosome (GO:0000228) in the cellular component category, and RNA binding (GO:0003723) in molecular function were most abundant GO terms (Figure 5A). In KEGG pathway enrichment analyses, viral carcinogenesis, ribosome, and transcriptional misregulation in cancer were significantly enriched (Figure 5B).

### 2.6. Mutation of the RKKR Impairs Nuclear Import of A23 Protein

The interaction between A23 protein and histone suggests the possibility of A23 protein possessing a nuclear localization signal (NLS) and being capable of entering the nucleus to perform its function. To predict the potential NLS, we utilized three NLS prediction software tools (Figure 6A–C). NLStradamus version r.9, with an adjusted prediction cutoff score of 0.2, identified an NLS at amino acids 123–126. The NucPred (https://nucpred.bioinfo.se/nucpred/, accessed on accessed on 7 May 2024) score for the amino acid sequence of A23 protein is 0.66. Sequences with a NucPred score ≥ 0.6, along with a PredictNLS prediction indicating the presence of a nuclear localization signal (NLS), have been shown to be 71% accurate with a coverage of 53% (https://nucpred.bioinfo.se/cgi-bin/single.cgi, accessed on 7 May 2024). Accordingly, the predicted NLS is RKKR, located at amino acids 123–126. Additionally, cNLS Mapper, with an adjusted cutoff score of 0.4, indicated that A23 protein localizes to both the nucleus and the cytoplasm. It identified RDRKKRSVEAFLDWMDTFGLRDSVPDRRKLD at amino acids 121–152 as a putative NLS. Consequently, the sequence RKKR at amino acids 123–126 within the MPXV A23 protein was consistently identified as an NLS.

Fluorescent inverted microscope analysis showed comparable results to transiently transfected cells (Figure 6D), and analysis of the nuclear to cytoplasmic fluorescence ratio (Figure 6E) confirmed the previous results. The WT-A23 protein shows a high degree of co-localization with the nuclear staining DAPI, and a higher nuclear vs. cytoplasmic ratio than the mutant A23_△RKKR_.

## 3. Materials and Methods

### 3.1. Cell Culture and Transfections

Human embryonic kidney (HEK) 293T cells were cultured in Dulbecco’s modified Eagle’s medium (DMEM) (Solarbio, Beijing, China) supplemented with 10% (*v*/*v*) fetal bovine serum (FBS) (ExCell Bio, Shanghai, China) at 37 °C and 5% CO_2_. HEK293T cells were passaged every 3 days. Upon reaching approximately 80–90% confluence, the cells were subcultured. During each passage, cells were split at a ratio of 1:3, with approximately 1 × 10^6^ cells seeded into each new culture vessel.

### 3.2. Plasmid Construction and Transfection

A 573 bp gene encoding MPXV A23R resolvase was codon-optimized and synthesized by TSINGKE (TsingKe Biotechnology, Beijing, China) based on the MPXV gene data (MPXV_USA_2022_MA001) published by NCBI (GenBank: ON563414.3). Primers were designed based on the MPXV-A23R gene sequence. The synthetic gene was inserted into the *EcoR* I and *Xho* I sites of pCAGGS-HA plasmid to obtain the pCAGGS-HA-A23R construct. The MPXV A23R coding sequence was amplified with PCR using the forward primer 5′-CCGGAATTCATGGAACCAGCCACCAGC-3′ (*EcoR* I site underlined) and the reverse primer 5′-CCGCTCGAGCATTTTGATATACGATATTACAAC-3′ (*Xho* I site underlined). The A23_△RKKR_-HA was amplified with PCR using the forward primer 5′-GCTATCGTGATAGCGTTGAAGCATTTCTGGATT-3′ and the reverse primer 5′-CAACGCTATCACGATAGCTATTACCGCTCATCA-3′, then transformed into *E. coli* for homologous recombination. The resulting clones were verified by DNA sequencing (capillary sequencing). According to the manufacturer’s instructions, when the cell density reached 85%, the pCAGGS-HA and pCAGGS-HA-A23R plasmids were, respectively, transfected into HEK293T cells using jetPRIME^®^ Transfection Reagent (Polyplus Transfection, Strasbourg, France) for a duration of 24 h. Subsequently, a subset of the transfected cells was lysed in Trizol (Vazyme, Nanjing, China) for RNA extraction and quantitative real-time RT-PCR (qPCR), while the remaining cells were lysed in RIPA buffer (Solarbio, Beijing, China) for further Western blot analyses and co-immunoprecipitation (co-IP).

### 3.3. RNA-Seq and Data Processing

The total RNA was extracted from the transfected HEK293T cells using a TransZol Up RNA Kit (Transgen, Beijing, China) according to the manufacturer’s instructions. Purified RNA was qualified and quantified using a Qubit^®^ 2.0 fluorimeter (Life Technologies, South San Francisco, CA, USA) and 2100 Bioanalyzer (Agilent Technologies, Santa Clara, CA, USA). The mRNA enrichment library was prepared. The cDNA libraries were generated using a NEBNext^®^ Ultra™ RNA Library Pre Kit for Illumina^®^ (NEB, Ipswich, MA, USA) and sequenced on an Illumina NovaSeq 6000 (Illumina, San Diego, CA, USA). The sequence coverage was 94.97% (Control) and 94.55% (A23R). The read length was 150 bp, and the end was paired end.

The RNA-seq analysis was performed by Novogene Biotech. The raw sequences were quality controlled and filtered using fastp software (v0.23.1) [20]. The high-quality clean reads were aligned to the reference genome of *Homo sapiens* GRCh38 using Hisat2 (v2.0.5) [21] with default parameters. The gene expression quantification was calculated by FPKM (fragments per kilobase of transcript per million fragments mapped) values by using featureCounts (v1.5.0-p3) [22]. The differential expression analysis was performed using the edgeR R package (v3.22.5) [23] with Padj ≤ 0.05 and |log2 (FC)| ≥ 1 as the difference significance criterion. GO and KEGG enrichment analyses were implemented by the clusterProfiler R package (v3.8.1) [24] with screening criteria of *p* < 0.05. A heatmap was plotted by using https://www.bioinformatics.com.cn (accessed on 6 May 2024), an online platform for data analysis and visualization.

### 3.4. Real-Time Quantitative Reverse Transcription PCR (qRT-PCR)

To confirm the reliability of the RNA-Seq data, twelve differentially expressed genes (DEGs) identified from the transcriptomic analysis were selected for qRT-PCR analysis. Briefly, cDNA was generated using Hifair^®^ V one-step RT-gDNA digestion SuperMix for qPCR (Yeasen, Shanghai, China) and qRT-PCR was carried out in triplicate on an Applied Biosystems 7500 Real-Time PCR Instrument (Life Technologies, USA). The qRT-PCR conditions were as follows: a pre-incubation (95 °C, 30 s), 40 amplification cycles (95 °C, 10 s; 60 °C, 30 s), and a final extension (95 °C, 15 s; 60 °C, 60 s). Primers used in this research were listed in the Appendix A, and β-actin was chosen as the internal reference. The relative expression level of each gene was calculated using the 2^−ΔΔCT^ method. The primers used are listed in Table 4.

### 3.5. LC-MS/MS and Data Processing

HEK293T cells transfected with pCAGGS-HA-A23R/control vector were lysed in RIPA buffer (Solarbio, Beijing, China) containing protease and phosphatase inhibitors. Cell lysates were incubated with anti-HA-agarose beads (Sigma Aldrich, Missouri, Germany) at 4 °C overnight. The beads were subsequently collected by centrifugation and washed with PBS prior to LC-MS/MS.

LC-MS/MS experiments and analysis were performed by Oebiotech. In brief, the proteins bound to the beads were denatured with 8 M urea in 50 mM NH_4_HCO_3_. The protein samples were sequentially reduced with dithiothreitol (DTT) (final concentration, 10 mM) at 55 °C for 30 min, alkylated with iodoacetamide (final concentration, 15 mM), enzymatically hydrolyzed with trypsin overnight, and desalted with a trap column, 100 μm × 20 mm (RP-C18, Thermo Inc, Waltham, MA, USA). Peptides were analyzed on a Q Exactive-HF mass spectrometer (Thermo Scientific, Waltham, MA, USA) coupled to an Easy-nLC 1200 (Thermo Scientific, USA). The raw data were searched against UniPort human database through Proteome Discover (v2.4). Precursor and fragment mass tolerance were set to 10 ppm and 0.6 Da, respectively. The local false discovery rate (FDR) was 1.0% at peptides with a maximum of two missed cleavage sites. The following settings were selected: Met-Loss, acetylation (protein N-term), deamidation (NQ), and oxidation (M) were selected for dynamic modifications and a fixed carbamidomethyl cysteine. Proteins with a ratio of abundance (A23R group/Control group) > 4 are considered to interact with A23 proteins. The LC-MS/MS data pathway analysis with the pCAGGS-HA and pCAGGS-HA-A23R plasmid-transfected cells was conducted using the DAVID bioinformatics resource (https://david.ncifcrf.gov/, accessed on 2 December 2023). The GO and KEGG pathways of *p* values less than 0.05 and 0.01 were considered significant and highly significant, respectively.

### 3.6. Western Blot Analyses and Immunofluorescence

Western blot analyses were performed using 12% sodium dodecyl sulfate–polyacrylamide gel electrophoresis (SDS-PAGE). The protein samples were electro-transferred to a 0.22 μm PVDF membrane (Millipore, Burlington, MA, USA). Membranes were blocked with 5% (*w*/*v*) skim milk-TBST at room temperature for 1 h and incubated overnight with antibodies against HA-tag, followed by incubation with corresponding horseradish peroxidase (HRP) conjugated antibodies (Proteintech, Wuhan, China). The signal was detected using a Chemic DOC imaging system (BioRad, Hercules, CA, USA).

HEK293T cells were grown on coverslips to 40–50% confluence, transfected with plasmids expressing A23-HA and A23_△RKKR_-HA, and incubated for 24 h. Cells were fixed with 4% paraformaldehyde, permeabilized with methanol, and washed three times with PBS. The fixed cells were stained with antibodies against HA-tag. Samples were sequentially incubated with fluorescent secondary antibody (Beyotime, Shanghai, China) and DAPI (Beyotime, Nantong, China) and visualized using the fluorescent inverted microscope.

Glass slides were mounted on the microscope stage and images were recorded through a 63× objective using a Nikon TI-S fluorescent inverted microscope.

DAPI was acquired through a 385–470 nm band pass filter using 5% of the UV laser intensity; HA was acquired through a 505–530 nm band pass filter using 5% of the 488 nm laser intensity. Single plane images were exported and analyzed with Image J (Version 1.54j). In order to assess the nuclear to cytoplasmic fluorescence ratio, a mask in the DAPI fluorescence image was used to identify the nucleus, and a mask in the HA fluorescence image was used to identify the whole cell. HA nuclear fluorescence was collected for all the cells analyzed and subtracted to the total cell fluorescence to generate cytoplasmic fluorescence. For each single cell analyzed the nuclear to cytoplasmic fluorescence ratio was calculated by dividing the nuclear HA fluorescence by the cytoplasmic HA fluorescence.

### 3.7. Prediction of the Nuclear Location Signal

The nuclear location signal was identified through NLStradamus [25] (http://www.moseslab.csb.utoronto.ca/NLStradamus/, accessed on 7 May 2024), NucPred [26] (https://nucpred.bioinfo.se/cgi-bin/single.cgi, accessed on 7 May 2024) and cNLS Mapper [27,28,29] (https://nls-mapper.iab.keio.ac.jp/cgi-bin/NLS_Mapper_form.cgi, accessed on 7 May 2024).

## 4. Discussion

Over the last two years, MPXV has emerged as a zoonotic disease with significant implications for global public health. While current case data suggest a decline in the frequency of MPXV outbreaks, the genome dynamic of DNA viruses poses a continued risk for transmission and evolution within vulnerable populations. With over 100,000 reported infections worldwide and the absence of a specific treatment, there is an urgent need to investigate the pathogenic mechanisms of MPXV [30]. The HJ resolvase demonstrates notable conservation in poxviruses and shows potential as a therapeutic target. Given the above reasons, we explored the characteristics and functions of MPXV A23 protein through RNA-seq and LC-MS/MS. In this study, a total of 648 DEGs were identified, comprising 314 upregulated genes and 334 downregulated genes, along with 115 cellular proteins interacting with MPXV A23 protein.

The poxviruses control the expression of the early, intermediate, and late genes through stage-specific transcription factors [31,32]. It was previously reported that HJ resolvase is predominantly expressed during the late stages of infection and is essential for the late-stage DNA replication and processing [33,34,35]. Garcia et al. showed that the suppression of VACV A22 resolvase resulted in a reduction in viral DNA replication at later stages, disruption of DNA concatemer resolution, and interruption of virion production [34]. A further study has indicated that this enzyme displays a broader range of substrate specificity compared to its closest relatives from the RNase H superfamily [35]. According to LC-MS/MS analysis, we identified a substantial quantity of ribosomal proteins interacting with MPXV A23 protein. These are, in GO terms, related to cytoplasmic translation (GO:0002181), cytosolic ribosome (GO:0022626), and translation (GO:0006412), including RPL10A, RPS26, RPS28, RPS15A, RPS16, RPL14, RPL24, RPL35, RPS2, RPL29, RPS11, and RPL17. This observation indicated the potential for A23 protein to exploit ribosomes for facilitating viral transcription, which is consistent with previous studies.

Poxviruses are known for their incredible self-sufficiency and intricate strategies to evade the host immune system. Remarkably, the innate immune response of the infected cells appeared virtually blind to the overwhelming infection of poxviruses [36]. The translational machinery employed by the virus enhances its replication efficiency and hinders the synthesis of host proteins, thereby contributing to the suppression of the host’s immune response [37]. This suggests that MPXV A23 protein serves various roles within the host cytoplasm. In addition to facilitating the accurate replication and repair of the viral genome, this protein collaborates with other viral proteins to inhibit host antiviral immune responses.

Ensuring the expression of certain host proteins is crucial for viral replication. Consequently, viruses have evolved mechanisms to bypass host shutoff by selectively synthesizing subsets of host proteins [38]. Understanding the mechanisms of selective host protein synthesis during host shutoff not only provides deeper insights into fundamental aspects of viral replication but also reveals novel strategies for combating viral diseases. For instance, VACV selectively synthesizes oxidatively phosphorylated proteins, as evidenced by significantly reduced viral replication when compound-induced oxidative phosphorylation is impaired during VACV infection [39]. This highlights the importance of oxidative phosphorylation in its lifecycle. We observed the enrichment of a subset of differentially expressed genes (DEGs) in cellular energy metabolic pathways, such as oxygen binding (GO:0019825), oxygen transporter activity (GO:0005344), oxygen transport (GO:0015671), and arachidonic acid and linoleic acid metabolism. However, it should be noted that the possibility of virus-induced inflammation and immune responses leading to increased cellular energy demands cannot be ruled out, warranting further investigation and confirmation.

Furthermore, previous studies have demonstrated a notable reduction in host cell mRNA levels after poxvirus infection, attributed to viral interference with various steps of host cell gene expression [40,41,42]. For instance, VACV utilizes the collaborative activity of decapping enzymes D9 and D10, along with the cellular nuclease XRN1, to degrade host mRNA [43,44,45]. Additionally, VACV infection impedes host transcription [41]. Sivan et al. observed that the knockdown of nucleoprotein 62 significantly blocked viral morphogenesis, underscoring the reliance of poxviruses on nuclear processes [46]. Here, the nuclear location signal (NLS) of the MPXV A23 protein was identified through bioinformatics analysis, revealing a sequence of RKKR at amino acids 123–126. Interactions were also observed between the A23 protein and histones, including H2AC4, H2AZ1, H2AX, H2AC18, H3C1, and H4C1 in nucleosome (GO:0000786). These results suggested that the MPXV A23 protein has the ability to translocate to the nucleus and potentially modulate the expression of host genes. Immunofluorescence experiments also showed that A23 protein is mainly localized in the nucleus, which provides a prerequisite for its interaction with histone proteins in the nucleus and the exercise of its corresponding biological functions.

Notably, our transcriptome data demonstrated that a significant number of DEGs were implicated in molecular pathways commonly associated with cancer, which mainly involved DNA damage repair and cell apoptosis. Apoptosis is a self-regulated cellular death mechanism that is activated by diverse stimuli to ensure tissue homeostasis and eliminate aberrant or infected cells [47]. As cell apoptosis plays a crucial role in the host’s antiviral defense, poxvirus has evolved multiple strategies to manipulate and disrupt apoptotic processes [48]. Our understanding of the apoptotic response triggered by MPXV infection is still limited, while related poxviruses exhibit significant anti-apoptotic abilities. For example, cytokine response regulator (CrmA) is one of the most potent anti-apoptotic proteins, which inhibits the activity of cysteine proteases (e.g., Caspase-1, Caspase-8, and Caspase-3) to disrupt apoptotic signaling [49,50]. Additionally, poxviruses can hinder the binding of cytokines, like TNF, to their receptors using decoy receptors, thereby reducing inflammation and cell apoptosis simultaneously [51]. VACV M1 anchor protein (ANK) interacts with apoptotic bodies, inhibiting the processing of protease-9 and downstream protease-3 [52]. Interestingly, poxviruses can exploit the cellular mechanisms for disposing of apoptotic bodies, facilitating intercellular spread through a process known as apoptosis simulation [53]. Alkhalil et al. employed GeneChip rhesus macaque genome microarrays to investigate the impact of MPXV infection on *Macaca mulatta* kidney epithelial cells (MK2), revealing that MPXV, similar to other poxviruses, demonstrates an anti-apoptotic tendency [54]. Our data indicated that MPXV A23 protein inhibits cell apoptosis at the transcriptional level, but it is unclear if the A23 protein directly regulates host cell apoptosis or if it is part of the host’s antiviral response. Further work is needed to validate and examine the role of the A23 protein in cell apoptosis. Collectively, our findings highlight the potential role of MPXV A23 protein in regulating host gene expression and modulating the host immune response.

## 5. Conclusions

The integration of transcriptomic and proteomic analyses facilitates the identification of gene regulatory networks and protein–protein interaction networks, thereby enhancing our comprehension of virus–host interactions (Figure 7, Table 5). The results presented here highlight the crucial role of the monkeypox virus A23 protein in the virus lifecycle. In addition to its involvement in late-stage DNA replication and morphogenesis, this protein may also play a role in (1) suppressing host antiviral immune responses and (2) regulating host gene expression in the cell nucleus.

## Figures and Tables

**Figure 1 ijms-25-08678-f001:**
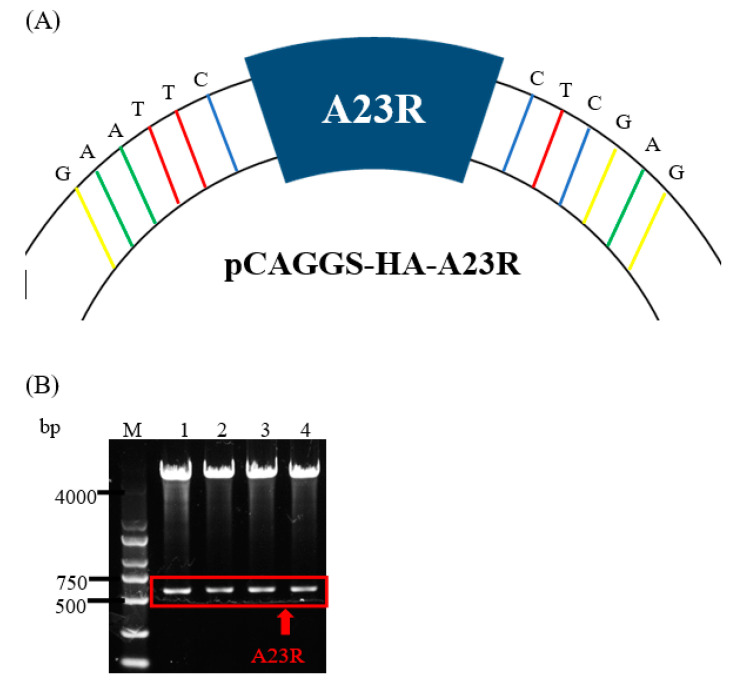
Construction and expression of recombinant A23R in HEK293T cells. (**A**) Model of a cloned fragment. (**B**) Double digestion of the pCAGGS−HA−A23R plasmid. Lane 1–4: A23R recombinants were digested by *EcoR* I and *Xho* I. (**C**) Expression of recombinant A23R in HEK293T cells, followed by immunoblot analysis using HA−tag antibodies or anti β−actin antibodies. Lane 1: Transfected with pCAGGS−HA. Lane 2: Transfected with pCAGGS−HA−A23R. (**D**) Western blot analysis of the Co−IP sample. Flow Through: the WCL after anti−HA incubate.

**Figure 2 ijms-25-08678-f002:**
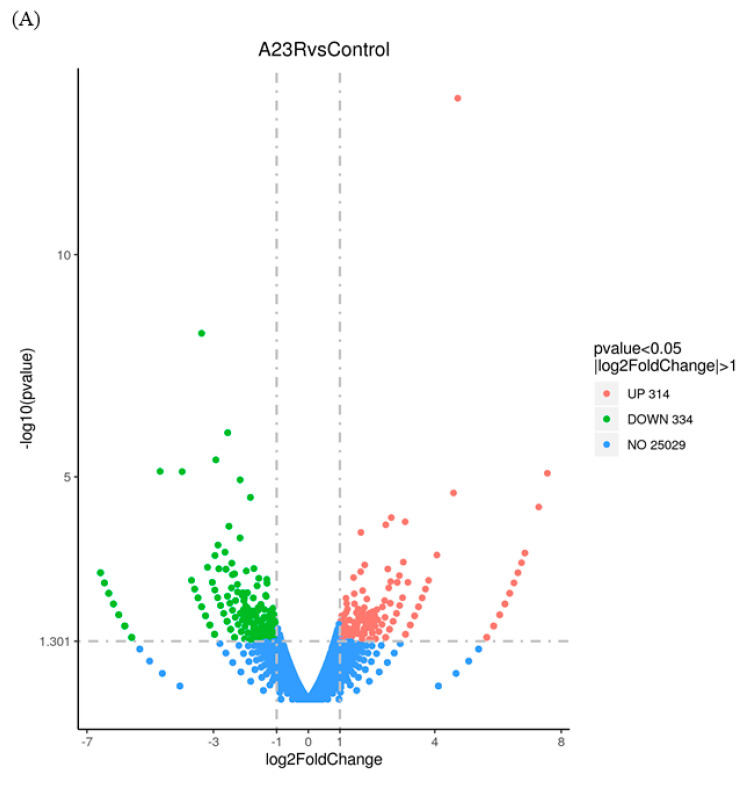
Identification of differentially expressed genes (DEGs). (**A**) The volcano diagram of DEGs. The horizontal axis represents the fold change in gene expression between the experimental and control groups (log_2_FoldChange). The vertical axis represents the significance of the DEGs between the experimental and control groups (−log_10_padj or −log_10_pvalue). Up−regulated genes are shown as red dots. Down−regulated genes are shown as green dots. Blue dots indicate no statistically significant genes (NO 25029). Threshold lines for DEGs screening criteria are indicated by blue dashed lines. (**B**) Heatmap of DEGs with length, type, and chr. The horizontal coordinate represents the sample name. The vertical coordinates on the left represent the cluster analysis. The vertical coordinates on the right represent length/type/chr. The heatmap specifies the length of each gene (length), categorizes its functions (type), and determines its position in the chromosome (chr). The red color in the middle of the heatmap represents high expression, and the green color represents low expression.

**Figure 3 ijms-25-08678-f003:**
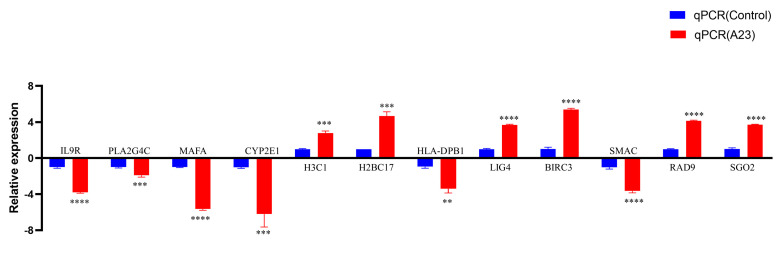
RT−qPCR verified the expression of seven genes. We examined the gene expression levels of *IL9R, PLA2G4C, MAFA, CYP2E1, H3C1, H2BC17, HLA−DPB1 LIG4, BIRC3, SMAC, RAD9*, and *SGO2* by RT−qPCR in HEK293T cells after the expression of A23 protein. RNA expression levels in each system were normalized to β−actin. The error bars indicate the SD of repeated RT−qPCR. All experiments were conducted in−dependently, at least three times. Statistical significance is indicated by ** *p* < 0.01, *** *p* < 0.001, **** *p* < 0.0001.

**Figure 4 ijms-25-08678-f004:**
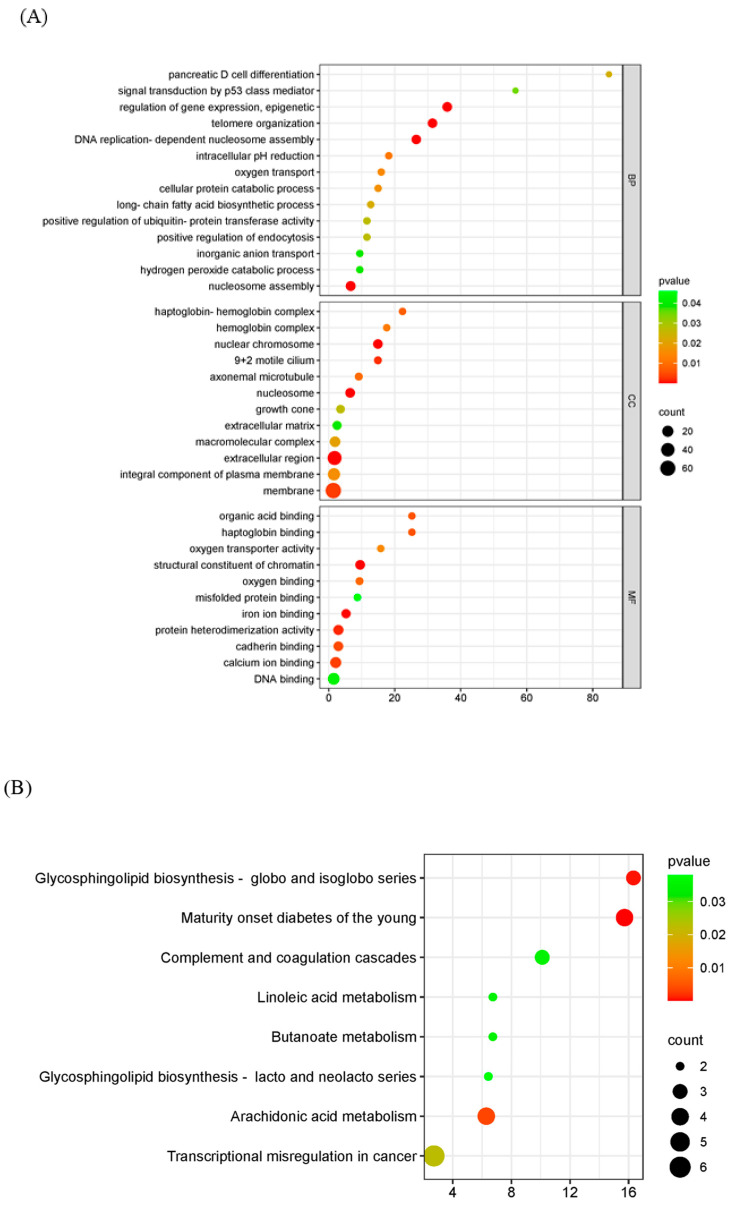
Functional enrichment analysis of DEGs. The scatter plot of GO (**A**) and KEGG (**B**) analysis. The vertical axis represents the top thirty terms with the most significance. The horizontal axis represents the gene ratio. Count: the number of DEGs. Gene ratio: the ratio of DEG number to background gene number. *p*-value: indicators of the significance of the term; the smaller the *p*-value, the more significant the term.

**Figure 5 ijms-25-08678-f005:**
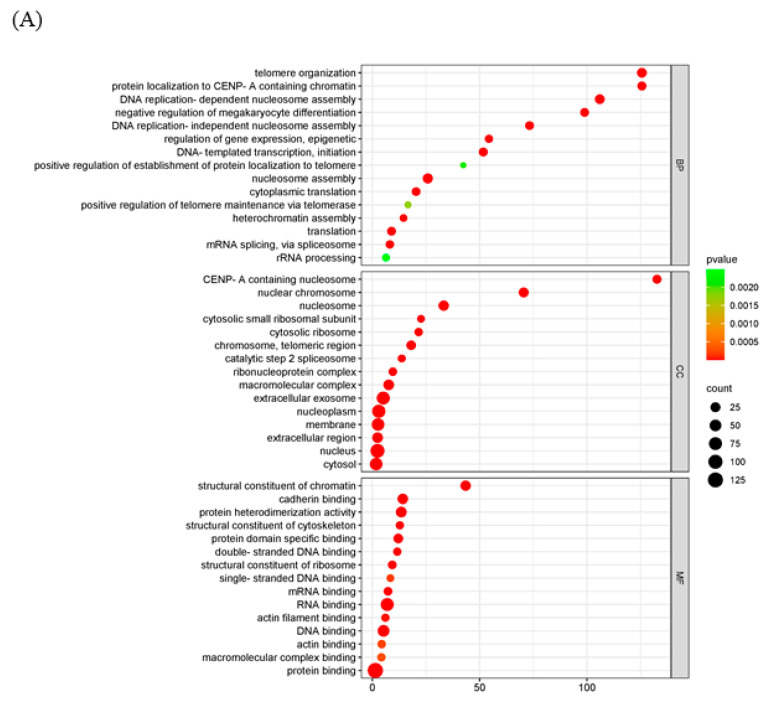
Functional enrichment analysis of the interacting proteins. The scatter plot of GO (**A**) and KEGG (**B**) analysis. The vertical axis represents the top thirty terms with the most significance. The horizontal axis represents the ratio. Count: the number of proteins. Gene ratio: the ratio of protein number to background protein number. *p*-value: indicators of the significance of the term, the smaller the *p*-value, the more significant the term.

**Figure 6 ijms-25-08678-f006:**
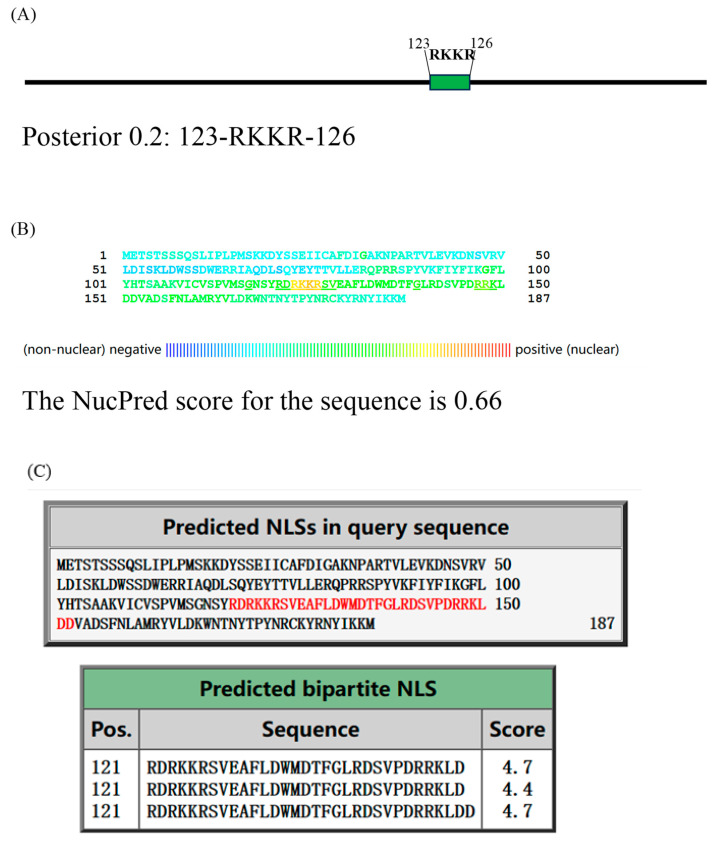
Mutation of the RKKR impairs nuclear import of A23 protein. The nuclear location signal in A23 protein was identified through NLStradamus (**A**), NucPred (**B**), and cNLS Mapper (**C**). HEK293T cells were transiently transfected with plasmic coding for HA-A23 and the A23_△RKKR_. (**D**) Transiently transfected cells were fixed, stained with DAPI, and analyzed by fluorescent Inverted microscope. (**E**) Nuclear localization of A23 protein and mutants was assessed in transiently transfected cells as a ratio of nuclear to cytoplasmic fluorescence using the Image J software version 1.54j. Data are from n = 4 fluorescent cells analyzed. ** *p* < 0.05 difference from WT-A23 transfected cells. Scale bar = 50 µm.

**Figure 7 ijms-25-08678-f007:**
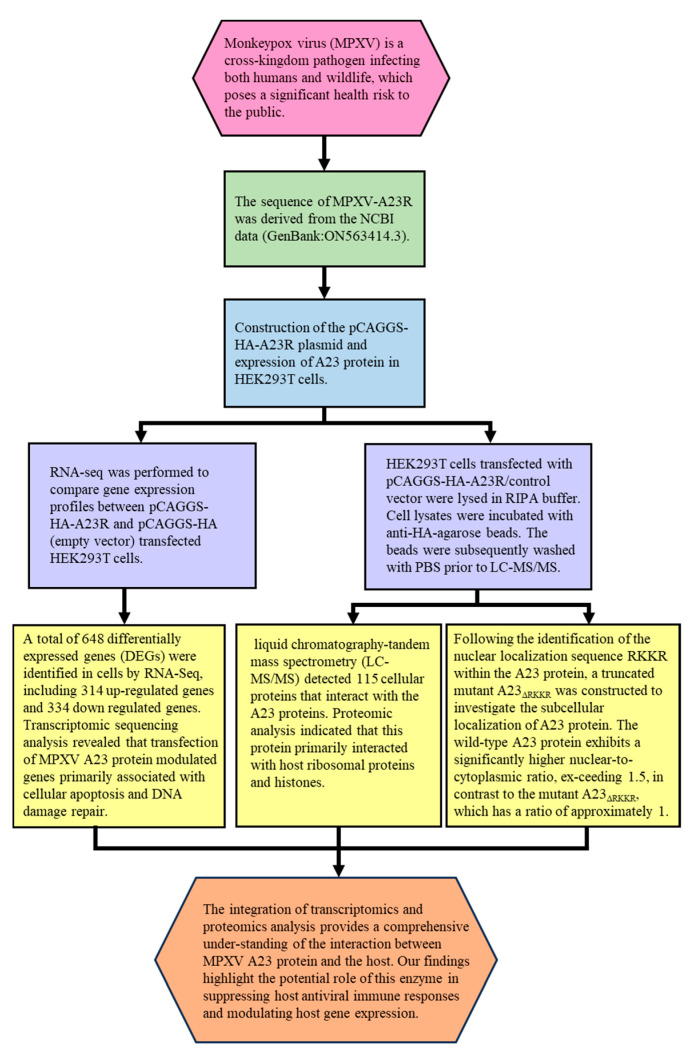
A combined transcriptomic and proteomic analysis of monkeypox virus A23 protein on HEK293T cells.

**Table 1 ijms-25-08678-t001:** Transcriptome sequencing evaluation.

Sample	Clean Reads	GC pct	Q20%	Q30%	Error Rate	Total Map	Unique Map	Multi Map
A23R	6.67G	48.55%	95.93%	89.62%	0.03%	94.55%	92.07%	2.48%
Control	6.47G	49.7%	96.28%	90.41%	0.03%	94.97%	92.49%	2.48%

Clean reads: the number of reads after raw data filtering; GC pct: G and C as a percentage of the four bases in clean reads; Q20: percentage of total bases with Phred values greater than 20; Q30: percentage of total bases with Phred values greater than 30; Error rate: sequencing error rate for the data; Total map: number of reads matched to the genome and their percentages; Unique map: number and percentage of reads matched to a unique position in the reference genome; Multi map: number and percentage of reads matched to multiple positions in the reference genome.

**Table 2 ijms-25-08678-t002:** The part list of RNA-Seq data.

Gene_id	A23R	Control	log2FoldChange	*p* Value	Gene_Name
ENSG00000124334	1.022537168	8.815369231	−2.964356163	0.034866602	*IL9R*
ENSG00000105499	3.06400515	13.713761	−2.118526375	0.030348148	*PLA2G4C*
ENSG00000182759	41.8522638	98.94567829	−1.238935084	0.013213222	*MAFA*
ENSG00000130649	6.126221523	21.5511823	−1.794423926	0.019137366	*CYP2E1*
ENSG00000275714	9.188448457	1.39 × 10^−17^	6.219344696	0.007243414	*H3C1*
ENSG00000274641	11.22993507	1.957591471	2.445440274	0.041032031	*H2BC17*
ENSG00000223865	34.7070525	81.31149061	−1.22536275	0.018875718	*HLA-DPB1*
ENSG00000174405	206.192132	94.04729284	1.131451016	0.012902837	*LIG4*
ENSG00000023445	20.4166311	5.876330133	1.774662417	0.033910222	*BIRC3*
ENSG00000284934	2.043272661	10.77472654	−2.331102932	0.041032031	*SMAC*
ENSG00000151164	24.49960836	2.937280525	3.00650683	0.000829981	*RAD9B*
ENSG00000163535	702.2739732	337.9868831	1.054779239	0.013382118	*SGO2*

**Table 3 ijms-25-08678-t003:** The part list of co-IP products.

Accession	Protein_Name	Score Sequest HT	Abundance: A23R	Abundance: Control	Ratio of Abundance (A23R/Control)
P04908	H2AC4	212.66	1.3 × 10^8^		
P0C0S5	H2AZ1	114.72	26,609,078		
P50914	RPL14	9.07	1,190,588		
P62244	RPS15A	8.62	12,072,808		
P47914	RPL29	6.38	3,892,084		
P62854	RPS26	4.09	4,662,978		
P18621	RPL17	1.65	5,741,302		
P62249	RPS16	7.63	5,723,261	30,704	186.4011
P42766	RPL35	10.51	13,601,984	87,385.85	155.6543
P15880	RPS2	8.85	3,162,923	51,026.81	61.98551
P83731	RPL24	3.3	3,176,778	51,312.03	61.91098
P62906	RPL10A	7.78	6,333,896	596,794.5	10.6132
P68431	H3C1	84.69	5.46 × 10^8^	51,672,486	10.56361
Q6FI13	H2AC18	190.38	1.42 × 10^9^	1.53 × 10^8^	9.29268
P16104	H2AX	197.19	16,754,252	2,945,901	5.687309
P62805	H4C1	141.17	1.83 × 10^9^	3.45 × 10^8^	5.314483
P62280	RPS11	21.92	22,282,617	5,319,779	4.188636
P62857	RPS28	17.31	11,419,641	2,801,809	4.07581

**Table 4 ijms-25-08678-t004:** Primers used for plasmid construction and real-time PCR.

Primer	Sequence (5′→3′)	Usage
A23R-F	5′-CCGGAATTCATGGAACCAGCCACCAGC-3′	pCAGGS-HA-A23R
A23R-R	5′-CCGCTCGAGCATTTTGATATACGATATTACAAC-3′	
A23R_△RKKR_-F	5′-GCTATCGTGATAGCGTTGAAGCATTTCTGGATT-3′	A23R_△RKKR_
A23R_△RKKR_-R	5′-CAACGCTATCACGATAGCTATTACCGCTCATCA-3′	
IL9R-F	5′-CGTGCCCTCTCCAGCGATGTTCT-3′	qRT-PCR
IL9R-R	5′-GACGCGCTGGGCCACAAGTG-3′	
PLA2G4C-F	5′-CCTTGAGTTCACCTTGGCTGTCCTAA-3′	qRT-PCR
PLA2G4C-R	5′-AAGGAGCAGTGGAAGGCATTGGTC-3′	
MAFA-F	5′-TCCTTCGTTCTCTTCTCAGCC-3′	qRT-PCR
MAFA-R	5′-AAAGAAGGGGCTTCCTCCAAG-3′	
CYP2E1-F	5′-CACAGTCGTAGTGCCAACTCT-3′	qRT-PCR
CYP2E1-R	5′-CACACACTCGTTTTCCTGTGGA-3′	
H3C1-F	5′-GTGTTCCGCTGTGCTGTTTT-3′	qRT-PCR
H3C1-R	5′-TAGCGGTGGGGCTTTTTCAC-3′	
H2BC17-F	5′-TACAACAAGCGCTCGACCAT-3′	qRT-PCR
H2BC17-R	5′-AGCTGCGAGAGCTCACTTG-3′	
HLA DPB1-F	5′-CAGCTCTTTTCATTTTGCCATCC-3′	qRT-PCR
HLA DPB1-R	5′-CTGGAAAAGGTAATTCTCTGGAGTG-3′	

**Table 5 ijms-25-08678-t005:** Clinical findings of a combined transcriptomic and proteomic analysis of monkey pox virus A23 protein on HEK293T cells.

Assays	Methods	Results
Plasmid construction and Western blot analyses of A23 protein	A 573 bp gene encoding MPXV A23R resolvase was codon-optimized and synthesized by TSINGKE (TsingKe Biotechnology, China) based on the MPXV gene data (MPXV_USA_2022_MA001) published by NCBI (GenBank: ON563414.3). Primers were designed based on the MPXV-A23R gene sequence. The synthetic gene was inserted into the EcoR I and Xho I sites of pCAGGS-HA plasmid to obtain the pCAGGS-HA-A23R construct. The MPXV A23R coding sequence was amplified with PCR, then transformed into *E. coli* for homologous recombination. The resulting clones were verified by DNA sequencing (capillary sequencing).	The pCAGGS-HA-A23R plasmids were successfully constructed and expressed in HEK293T (Figure 1B,C).
Transcriptome sequencing evaluation	RNA-seq was performed to compare gene expression profiles between pCAGGS-HA-A23R- and pCAGGS-HA (empty vector)-transfected HEK293T cells. The raw sequences were quality controlled and filtered using fastp software (v0.23.1). The high-quality clean reads were aligned to the reference genome of Homo sapiens GRCh38 using Hisat2 (v2.0.5) with default parameters.	The transcriptome sequencing data were appropriate and reliable for further analysis.
Identification of DEGs	The gene expression quantification was calculated by FPKM (fragments per kilobase of transcript per million fragments mapped) values using featureCounts (v1.5.0-p3). The differential expression analysis was performed using the edgeR R package (v3.22.5) [23] with Padj ≤ 0.05 and |log2 (FC)| ≥ 1 as the difference significance criterion.	When compared with cells transfected with empty vectors, a total of 648 DEGs were screened in HEK293T cells transfected with pCAGGS-HA-A23R, of which 314 genes were upregulated, and 334 genes were downregulated (Figure 2 and Figure 3).
Functional and pathway enrichment analysis of DEGs	GO and KEGG enrichment analyses were implemented by the clusterProfiler R package (v3.8.1) with screening criteria of *p* < 0.05. A heatmap was plotted using https://www.bioinformatics.com.cn, an online platform for data analysis and visualization.	In the GO enrichment analysis and KEGG pathway enrichment analyses, transcriptional misregulation in cancer, arachidonic acid metabolism, and linoleic acid metabolism were statistically significantly enriched (Figure 4).
Interaction of MPXV A23R	HEK293T cells transfected with pCAGGS-HA-A23R/control vector were lysed in RIPA buffer. Cell lysates were incubated with anti-HA-agarose beads. The beads were subsequently washed with PBS prior to LC-MS/MS. According to the manufacturer’s instructions (Oebiotech, Shanghai, China). The samples were enzymatically digested into peptides, which were desalted and evaporated to dryness. Separation was performed using a Nano-HPLC liquid phase system (EASY-nLC1200). The enzymatic products were separated by capillary high-performance liquid chromatography. The mass spectrometry analysis was carried out with a Q-Exactive HF mass spectrometer (Thermo Scientific).	In the GO enrichment analysis and KEGG pathway enrichment analyses, viral carcinogenesis, ribosome and transcriptional misregulation in cancer were significantly enriched (Figure 5).
Mutation of the RKKR impairs nuclear import of A23 protein	Glass slides were mounted on the microscope stage and images were recorded through a 63×objective using a Nikon TI-S fluorescent inverted microscope. DAPI was acquired through a 385–470 nm band pass filter using 5% of the UV la-ser intensity; HA was acquired through a 505–530 nm band pass filter using 5% of the 488 nm laser intensity. For each single cell analyzed, the nuclear to cytoplasmic fluorescence ratio was calculated by dividing the nuclear HA fluorescence by the cytoplasmic HA fluorescence.	The A23 protein shows a high degree of co-localization with the nuclear staining DAPI, and a higher nuclear vs. cytoplasmic ratio than the mutant A23_△RKKR_ (Figure 6).

## Data Availability

The datasets used and/or analyzed in this study are available from the corresponding authors upon a reasonable request. This data can be found here: [NCBI/PRJNA1108403] and [ProteomeXchange/PXD052000].

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
