# Peer review of "A Combined Transcriptomic and Proteomic Analysis of Monkeypox Virus A23 Protein on HEK293T Cells"

_ijms, 2024, doi:10.3390/ijms25168678_

Round 1
Reviewer 1 Report
Comments and Suggestions for Authors
Yihao et al., conducted a transcriptomic and proteomic analysis to underpin the role of A23 protein of MPXV on Human cell line. The study is interesting and well supported extensive wet and dry-lab works and has potential to interest readers working on infectious diseases , host-virus interaction and Multi-omics dataset. To me, the manuscript has quite a few typos, which needed to be fix, including Italicising scientific names. Also, Methods section need some improvement to ensure reproducibility. Discussion section is well supported with lots of literatures, but the continuation from one paragraph to another is missing. It will be good, if Authors can re-visit the discussion and ensure the flow of their “story”. Few of my concerns are below -
Abstract –
Line 14 and line 16-17 are saying same things. Better to eliminate one, or shorten the first one.
Transition from line 71 to new paragraph is not clear. Add a linker sentence at Line 72.
Line 79 - Escherichia coli should be italic.
Line 93 – “RNA-Seq transcriptomics”, use either RNA-Seq or Transcriptomics.
Line 87-94 - The motivation of the study is not clear. How can transcriptome and proteome analysis help? What would be the implication?
Line 99 – need more details. How often cells were passaged? Number of passages?
Line 103 – 110 – Not clear if the primers were developed or adopted from literatures. Need to add those details / citations/ primer design info briefly.
Line – 111- “clones were verified by DNA sequencing.” – which type of sequence? Capillary or NGS?
Line 115 – sampling time, number/replicates, control info are not clear. Need to add to ensure reproducibility.
Line 118 – which library prepared? mRNA enrichment or rRNA depletion.
Line 120 – need sequence coverage, read length, single or paired end?
Line 123 – Homo sapiens will be Italic.
Line 140 – action or Actin?
Line 152 – “C18 columns” dimensions and supplier details missing.
Line 159 – the intermediate steps from Proteome discoverer and DAVID is missing. Which comparison were made? What fold changes were considered significant? What go into DAVID analysis? any parameters considered for DAVID? What was the significance level for GO analysis? – need to describe these to ensure reproducibility of the study.
Line 197 – “one 500 bp is A23R gene in gel” – in Fig 1B, the band size seems bigger than 500bp.
Line 237 – “Thermal polymerization map” can be “heatmap”.
Line 239 – “The vertical coordinates on the right represent gene names” – cannot see any gene names.
Fig 2B – describe “length”, “types” in caption.
Line 243-245 and Fig 3 – did authors perform any statistical test, to show if RNAseq and qPCR based expression correlates? It needs evidence to support their claim “ Further support the credibility of RNA-Seq”
Line 406 - Macaca mulatta will be Italic
Discussion: overall nice discussion, which supports the findings from RNAseq and LC-MS studies. I can see lots of binding GO terms in both analysis, which needed to expanded.
Line 346-348 – I cannot find any result supporting the interaction of MPXV A23 protein with other RP proteins.
Comments on the Quality of English Language
Minor editing of English language required. Need to fix typos and Scientific names.
Author Response
Dear Reviewer:
Many thanks for the insightful comments and advice concerning our manuscript entitled “A combined transcriptomic and proteomic analysis of monkeypox virus A23 protein on HEK293T cells (Manuscript ID: ijms-3098905)”. We appreciate you and the reviewers for your time and effort. Those comments are valuable and helpful for improving our work. We have carefully considered these comments and made corresponding revisions according to the comments. We have submitted our documents with tracked changes to highlight the revisions. The main correction and the point-by-point response to the reviewer’s comments are presented at the attachment.
Thank you for allowing us to resubmit a revised edition of the manuscript and we highly appreciate your time and consideration.
Lingbao Kong, Ph. D, Professor; Meifeng Li, Ph. D, lecturer
Institute of Pathogenic Microorganism
Jiangxi Agricultural University
Nanchang, Jiangxi Province 330045, P.R. China
Tel: + 86 791 83813459
Fax: +86 791 83828080.
E-mail: lingbaok@mail.jxau.edu.cn; meifengli77@jxau.edu.cn

Reviewer 2 Report
Comments and Suggestions for Authors
This manuscript demonstrates that the A23 protein of MPXV regulates apoptosis and DNA damage repair in host cells, and finds that the A23 protein of MPXV interacts with ribosomal proteins and histone proteins in the nucleus of the cell, which is helpful for the study of the inhibition of host antiviral immune response and the modulation of host gene expression, however, there are some questions that need to be answered by the authors, for example:
1. In Fig.2A, what is the NO 25029? I do not see any description shown in the content, please explain in detail.
2. Line 226-229: “Notably, a considerable portion of genes associated with cell apoptosis (e.g., LIG4, BIRC3, and SMAC) and DNA damage repair (e.g., RAD9B, LIG4, and SGO2), were significantly differently expressed”, I do not see the results in Fig. 2. Please provide the details.
3. Line 243-245: the authors analyzed randomly selected 7 DEGs (IL9R, PLA2G4C, MAFA, CYP2E1, H3C1, H2BC17, and HLA-DPB1) using qRT-PCR in experiment Fig. 3, and I would like to ask why the authors did not analyze these genes directly, which are related to apoptosis (e.g., LIG4, BIRC3, and SMAC) and DNA repair (e.g., RAD9B, LIG4, and SGO2), and please explain the reason.
4. Line 273-275: There is no Fig. 2D and Table 3 ain this manuscript, please revise them.
5. Line 276~280: LC-MS/MS analysis showed that a large number of ribosomal proteins (e.g., RPL10A, RPS26, and RPS28) and RNA polymerases (e.g., POLR2H, and POLR2G) can interact with the MPXV A23 protein, which can promote the replication of viruses, please revise them. I would like to ask where the LC-MS/MS analysis result graph is, please revise it.
6. Iine: 276~280, a large number of histone proteins (e.g., H2AC4, H2AZ1, and H2AX) were analyzed, and it was stated that MPXV A23 protein could regulate the host gene expression in the nucleus, please tell me how to get this conclusion, and whether there is a result graph to prove it, please explain in detail.
7. Line 303-305: “NucPred score >= 0.6, along with a PredictNLS prediction indicating the presence of a nuclear localization signal (NLS), have been shown to be 71% accurate with a coverage of 53%”. Where did these values come from? Please add the relevant data and explain in detail.
Comments on the Quality of English LanguageNeed to be extensive revised
Author Response

(The authors gave the same response as above.)

Reviewer 3 Report
Comments and Suggestions for Authors
This review manuscript has scientific merit that might benefit readers, but some revisions still need to be included. Therefore, we recommend that the authors modify it.
1. The authors have mentioned the background, methods, results, and conclusion parts in the abstract. We suggest the authors remove these headings and make the paragraph descriptive paragraph without mentioning the headings.
2. The authors are suggested to mention the numerical values of results with standard units in the abstract. For example, Transcriptomic sequencing analysis, Proteomic analysis, and Immunofluorescence assay result values with standard unit numbers.
3. In the introduction, the authors need to change the name of Escherichia coli to italics because bacterial strains or plant names are always written in italics.
4. In section 2.4. Real-time quantitative reverse transcription PCR (qRT-PCR) and section 3.4. Functional and pathway enrichment analysis of DEGs:
"To validate the reliability of the RNA-Seq data, 7 differentially expressed genes (DEGs) were arbitrarily selected for qRT-PCR analysis."
"To clarify the functions of these DEGs, they were assigned to three main categories: biological process (14 terms), cellular component (12 terms), and molecular function (11 terms)." The authors need to rewrite the sentences as the structure is not professional.
5. In the discussion part, the authors mentioned: "Over the last two years, mpox has emerged as a zoonotic disease with significant implications for global public health." They mistakenly wrote mpox instead of MPXV. Please review this.
The last paragraph of the discussion must be revised: "In this study, we integrated transcriptomic and proteomic analyses to elucidate the interaction between the MPXV A23 protein and host cells. Our findings are consistent with previous reports of related poxviruses, highlighting the potential role of this enzyme in regulating host gene expression and modulating the host immune response. Further work is necessary to validate and explore the potential of this viral protein in antiviral therapeutics."
The discussion part must be shortened as it resembles a repetition of results. The authors should restructure the discussion part.
6. The first sentence of the conclusion section uses the word "elucidation" again, which is used multiple times in the manuscript. We suggest rewriting the sentence and presenting it with scientific justifications.
7. The authors are encouraged to create a schematic diagram comprehensively illustrating their research work. To be included in the results, this diagram should be titled "A combined transcriptomic and proteomic analysis of monkey-2 pox virus A23 protein on HEK293T cells."
Figure 3: A combined transcriptomic and proteomic analysis of monkey-2 pox virus A23 protein on HEK293T cells.
The illustration should show all the parts of the research article:
- Introduction
- Methods
- Results
- Conclusion
This illustration will give a short overview of the research and attract scientific readers to read this research work, improving readability and outreach.
8. The authors must create one table in the results section that demonstrates all the results of their respective research findings. This descriptive table will help make their research work more scientifically appealing.
Table 1: Clinical findings of a combined transcriptomic and proteomic analysis of monkey-2 pox virus A23 protein on HEK293T cells.
|
Assays |
Sequence used |
Prospective bands |
Results |
|
Plasmids construction and Western blot analyses of A23 protein |
|
|
|
|
Transcriptome sequencing evaluation |
|
|
|
|
Identification of DEGs |
|
|
|
|
Functional and pathway enrichment analysis of DEGs |
|
|
|
|
Interaction of MPXV A23R |
|
|
|
|
Mutation of the RKKR impairs nuclear import of A23 protein |
|
|
|
Comments on the Quality of English Language
The authors should rewrite a few sentences, especially the last paragraph of the discussion, as mentioned in the comments section.
Author Response

(The authors gave the same response as above.)

Round 2
Reviewer 2 Report
Comments and Suggestions for Authors
The revised manuscript is fine for me. No further comment on it.
Reviewer 3 Report
Comments and Suggestions for Authors
The authors have addressed all the comments in detail and made the suggested changes.
Best wishes.